# Nutrient Extraction Lowers Postprandial Glucose Response of Fruit in Adults with Obesity as well as Healthy Weight Adults

**DOI:** 10.3390/nu12030766

**Published:** 2020-03-14

**Authors:** Rabab Alkutbe, Kathy Redfern, Michael Jarvis, Gail Rees

**Affiliations:** School of Biomedical Sciences, University of Plymouth, Drake Circus, Plymouth PL4 8AA, UK; rabab.alkutbe@plymouth.ac.uk (R.A.); kathy.redfern@plymouth.ac.uk (K.R.); michael.jarvis@plymouth.ac.uk (M.J.)

**Keywords:** glycemic index, obesity, raspberry, passionfruit, postprandial

## Abstract

Fruit consumption is recommended as part of a healthy diet. However, consumption of fruit in the form of juice is positively associated with type 2 diabetes risk, possibly due to resulting hyperglycemia. In a recent study, fruit juice prepared by nutrient extraction, a process that retains the fiber component, was shown to elicit a favorable glycemic index (GI), compared to eating the fruit whole, in healthy weight adults. The current study expanded on this to include individuals with obesity, and assessed whether the nutrient extraction of seeded fruits reduced GI in a higher disease risk group. Nutrient extraction was shown to significantly lower GI, compared to eating fruit whole, in subjects with obesity (raspberry/mango: 25.43 ± 18.20 vs. 44.85 ± 20.18, *p* = 0.034 and passion fruit/mango (26.30 ± 25.72 vs. 42.56 ± 20.64, *p* = 0.044). Similar results were found in those of a healthy weight. In summary, the current study indicates that the nutrient-extraction of raspberries and passionfruit mixed with mango lowers the GI, not only in healthy weight individuals, but also in those with obesity, and supports further investigation into the potential for nutrient extraction to enable increased fruit intake without causing a high glycemic response.

## 1. Introduction

Fruit consumption has beneficial health effects that correlate with the decreased risk of several chronic diseases [1]. As a consequence, public health agencies consistently promote the protective health effects of fruit. However, fruit juice is known to cause a postprandial peak in blood glucose, and it is advised to limit juice intake due to this high glycemic index (GI). Consistent with the effect on GI, research from large cohort studies has shown an association between fruit juice consumption and an increased risk of type 2 diabetes (T2DM) [2,3]. It has been suggested that the decrease in fiber per serving of fruit juice compared with whole fruit may explain the increased risk of T2DM. 

Recommendations to increase fruit intake have increased the popularity of ‘nutrient-extraction’ blenders that homogenize whole fruit to create ‘smoothies’ without the removal of fiber. This form of processing is in contrast to traditional juicers that eliminate pulp, with most of the current GI-focused literature being based upon studies using fruit juice devoid of pulp. The impact on health of the consumption of fruit that has been nutrient-extracted, rather than juiced in a traditional manner, therefore remains unclear, and requires investigation for public health guidance. It is particularly important to understand the effect on the glycemic response in individuals susceptible to glucose intolerance, such those with obesity, as this population is more susceptible to T2DM onset [4]. These disease-susceptible individuals also make up an ever-increasing proportion of the population in high income countries, such as the UK (63%) [5].

Findings from an earlier study in young, healthy adults demonstrated that the consumption of mixed fruit following nutrient extraction resulted in a significant lowering of the GI, compared with the non-extracted (whole) fruit [6]. In contrast to the fruit mixture, the nutrient extraction of mango as a single fruit had no effect on the GI. These findings suggested, that in contrast to conventionally prepared fruit juice, fruit juice prepared by nutrient extraction in some cases elicits a more favorable postprandial glycemic response than whole fruit. 

Notably, even for high GI fruits like mango, nutrient extraction did not worsen the response. The mechanism responsible for this effect was unclear, and it was also unknown whether a similar response would be elicited in older or overweight sample groups.

The aim of the current study was to extend the investigation into the effect of nutrient extraction on the postprandial glycemic response to individuals with obesity. As nutrient extraction retains the fibrous pulp, it has been hypothesized that use of fruits with seeds will lower the glycemic response, as the fiber (and fats/proteins) from the seeds will be released and become available. This is compared to eating the whole fruit, where most seeds will remain intact. The effect of seeded fruit (raspberries and passionfruit) combined with mango, on the nutrient extraction effect, was therefore also studied. 

## 2. Materials and Methods

### 2.1. Participants

Participants considered as of healthy weight (BMI < 25 kg/m^2^) or obese (BMI > 30 kg/m^2^) were recruited to the study. Two groups of healthy weight participants were recruited for the completion of raspberry (*n* = 15) and passionfruit arms (*n* = 12), respectively. Participants with obesity completed both study arms (*n* = 12). There were three dropouts from the study, due to not completing all of the tests (two with healthy weight in the passionfruit arm, and one with obesity). Therefore, from the original 39, a total of 36 participants completed one or both arms of the study. All participants completed a health screening questionnaire. Exclusion criteria were pregnancy, age <18 years, fruit allergy, known diabetes, and use of medication known to interfere with glucose homeostasis or intestinal absorption. Written, informed consent was obtained from participants, and the study was approved by the Research Ethics Committee of the Faculty of Science and Engineering.

### 2.2. Materials/Processing

Two different seeded fruits were investigated: raspberries mixed with mango, and passionfruit mixed with mango. The raspberry arm consisted of 162 g raspberries and 114.2 g of mango in each test meal, and the passionfruit arm consisted of 150 g passionfruit and 114.2 g mango in each test meal.

Two test meals were provided for each study arm: (i) whole fruit or (ii) nutrient-extracted fruit, with both meals containing 25 g total sugar per serving (Figure 1). Total sugar was calculated from a high performance liquid chromatography (HPLC) analysis of raspberry, passionfruit and mango, with 12.5 g of total sugar coming from each of the two fruits used in each arm of the study. Meals were prepared on the morning of the test. Nutrient-extracted servings were processed freshly in a 600 W, 20,000 r.p.m. blender (Nutribullet 600, Nutribullet LLC, Pacoima, CA, USA) for 30 s with 125 mL water, transferred to a plastic cup, and sealed. Whole-fruit servings were cut into bite-sized pieces, transferred to a container, and consumed with 125 mL of water. The control arm of 25 g glucose was dissolved in 125 mL water. Test meals (glucose control, nutrient extracted or whole fruit) were consumed in a random order (Figure 1). 

### 2.3. Experimental Protocol

A crossover design was used for this study, with each participant serving as their own control. Participants consumed each test meal or control, with a minimum 2-day washout period between test days. Participants fasted for 12 h overnight, and avoided alcohol, caffeine and vigorous exercise in the 24 h preceding testing. Fasting glucose levels were obtained via a finger prick blood sample on arrival at 9 am (Accu-Check Advantage, Roche, Welwyn Garden City, UK). Testing began with the first oral contact with the test meal, which was then consumed over a 15-min period, with 125 mL of water for the whole fruit test. Postprandial blood glucose levels were determined at 15, 30, 45, 60, 75, 90, 105 and 120 min for each condition.

The glycemic index (GI) was calculated from the incremental area under the 120 min glucose response curve for each test meal. The incremental area under the curve for each test meal was expressed as the percentage of the mean area under the glucose control curve for the same subject. These values were used to calculate the GI values for each test meal using methods described by Brouns et al. [7].
(1)$$\begin{array}{c} \mathrm{GI}\;(\%)=\mathrm{Incremental}\;\mathrm{area}\;\mathrm{under}\;\mathrm{the}\;120\;\min\;\mathrm{glucose}\;\mathrm{response}\;\mathrm{curve}\;\mathrm{for}\;a\;25\;g\\ \mathrm{carbohydrate}\;\mathrm{equivalent}\;\mathrm{of}\;\mathrm{the}\;\mathrm{test}\;\mathrm{fruit}\times 100/\mathrm{incremental}\;\mathrm{area}\;\mathrm{under}\;\mathrm{the}\;120\;\min\\ \mathrm{glucose}\;\mathrm{response}\;\mathrm{curve}\;\mathrm{for}\;25\;g\;\mathrm{glucose}\;\mathrm{load} \end{array}$$

### 2.4. Statistical Analysis

Data were analyzed using the SPSS 23 Statistical software (IBM Corp, Armonk, NY, USA). All data were assessed for normal distribution of values. A one-way repeated-measures analysis of variance (ANOVA) was used to identify the significant differences of GI between test meals for all participants together, and for the different BMI groups. Post-hoc tests were conducted for each time point using the LSD method for multiple comparisons. In order to test the effect of sample size, Cohen’s (1988) criteria was conducted [8]. 

Data are reported as mean values ± SEM in figures, or mean with SD in text and tables, unless otherwise stated, and significant differences were considered at *p <* 0.05. 

## 3. Results

A total of 26 participants completed the raspberry and mango arm of the study (healthy weight *n* = 15, obese = 11) and 21 completed the passionfruit and mango arm of the study (healthy weight *n* = 10, obese *n* = 11). Table 1 details characteristics of all study participants. The number of participants in each group is slightly different, and so to test the effect of sample size a statistical analysis was conducted based on Cohen’s (1988) criteria. The effect size was very small (partial eta squared = 0.003), meaning that the difference in sample size had little effect on the results.

As expected, both healthy weight groups from each arm had a significantly lower weight, BMI, body fat percentage, waist circumference and fasting blood glucose, compared with the group with obesity. 

First, we wanted to determine the impact of nutrient extraction on the raspberry/mango (RM) and passionfruit/mango (PFM) arms in all participants when viewed as a single, combined study group. GI values for all test meals, representing the incremental area under the entire 120 min time course, can be seen in Figure 2a,b. For both RM and PFM test arms, nutrient extraction resulted in a significant reduction in GI. Figure 2a shows the GI results for the RM arm. The mean GI for the nutrient-extracted RM (36.6 ± 26.2) was significantly lower than the whole RM (52.8 ± 24.1; *p* < 0.019) and glucose control (100.0 ± 61.0; *p* < 0.001). Figure 2b illustrates the GI results for the PFM arm, and similarly shows the GI of nutrient-extracted PFM (32.1 ± 26.6) to be significantly lower than for the whole PFM (56.0 ± 34.0; *p* = 0.026). The mean GI for both nutrient-extracted and whole PFM fruit were significantly lower than glucose (100.0 ± 61.0; *p* < 0.001), as expected. 

Next, we examined the postprandial glucose responses after consumption of whole fruit, nutrient-extracted fruit and glucose (control). Results are shown in Figure 3a,b for RM and PFM arms, respectively, for all participants as a single study group. Consistent with the GI results (Figure 2), both RM and PFM nutrient-extracted arms showed a significant reduction in the postprandial glucose response, compared to whole fruit. In the RM arm, glucose and whole fruit peaked at 30 min, while the nutrient-extracted arm exhibited a slower rise, with the peak period occurring between 30 and 45 min. In the PFM arm, blood glucose peaked at 30 min for all three test meals. However, glucose responses in the nutrient-extracted PFM arm at 90 min declined below the pre-prandial level, and returned to the starting points by 105 and 120 min. As expected, the glucose meal in both arms elicited the greatest rise above pre-prandial levels, and then declined below pre-prandial levels at later times. 

Consistent with the results shown in Figure 3, post hoc analysis showed that following the consumption of nutrient-extracted RM, blood glucose levels were significantly lower than the whole fruit at 30 min (*p* = 0.042)) (Figure 3a). Similar differences were observed in blood glucose following consumption of the nutrient-extracted PFM fruit, which was significantly lower than the whole fruit at both 15 and 30 min (*p* = 0.046, *p* = 0.023 respectively) (Figure 3b).

In order to determine whether the effect of nutrient extraction on GI extended to individuals with obesity, the effect of BMI on the postprandial blood glucose response after the consumption of whole fruit and nutrient-extracted fruit was determined for the different BMI groups. The GI for each fruit arm for groups with healthy weight and with obesity is presented in Figure 4. 

As observed when examining the GI responses for all participants as a single group (Figure 2), significant reductions in GI values were observed in both participants, with obesity and healthy weight individuals when consuming the nutrient extracted fruit compared to whole fruit. This pattern was observed in both the raspberry-mango (Table 2) and passionfruit-mango arms (Table 3). 

There were some differences in the responses to the different fruit arms between those with a healthy weight and those with obesity. The mean GI for whole raspberry-mango was higher in the healthy weight group compared to the group with obesity (*p* = 0.036) (Table 2). The results for the nutrient extracted fruits show a comparable effect in both BMI groups with no significant difference between healthy weight and obese groups. In the passionfruit and mango arm, there were no significant differences in the GI values obtained for any of the tested meals between BMI groups, but there was a trend for higher GI responses in the healthy weight compared to the group with obesity (Table 3).

Figure 5a demonstrates the incremental changes in blood glucose for participants with a healthy weight, and with obesity, after their consumption of whole fruit, nutrient-extracted fruit and glucose (control) for the raspberry-mango group. The healthy weight participants’ blood glucose responses were significantly greater after the consumption of the whole RM than the NE meal at 30 min (*p* = 0.002). Similarly, the blood glucose responses for participants with obesity were significantly higher in the whole RM at the 15, and 120 min post-prandially than the nutrient-extracted meal (*p* = 0.014, 0.024, respectively) (Figure 5a). 

Figure 5b shows the blood glucose responses in the passionfruit and mango arm for the healthy weight participants and the participants with obesity. The blood glucose response for whole fruit for participants with a healthy weight was significantly greater than the nutrient-extracted meal at 30, and 45 min (*p* < 0.001, 0 = 0.015, respectively). Whereas this difference was only seen in the participants with obesity at 30 min (*p* = 0.028) (Figure 5b). 

A comparison in the incremental changes in blood glucose for participants with a healthy weight, and with obesity, after their consumption of whole fruit, nutrient-extracted fruit and glucose, was conducted. For the raspberry-mango arm (Figure 5a), the blood glucose responses for participants with obesity were significantly higher than those with a healthy weight for the glucose control: at 30, 45, 60, 105 min (*p* = 0.039, 0.010, 0.002, 0.05, respectively). Conversely, the healthy weight participants’ blood glucose responses were significantly greater than those who were obese at 15 min for the nutrient-extracted meal (*p* = 0.003) and at 15, 30, 105 and 120 min for the whole fruit meal (*p* = 0.042, 0.006, 0.05, 0.005, respectively). 

Similarly in the passionfruit-mango arm, the blood glucose response for participants with obesity was significantly greater than healthy weight participants for the glucose control at 45, 60, 75 and 90 min (*p* = 0.045, 0.05, 0.02, 0.043, respectively). In addition, the blood glucose response to the nutrient-extracted meal was significantly higher in the participants with obesity at 75 min (*p* = 0.044). There were no differences in blood glucose change observed between the healthy weight and obese groups after the consumption of the passionfruit-mango whole fruit meal. 

## 4. Discussion

In line with our hypothesis, the present study demonstrates that nutrient-extraction preparation of raspberries and passionfruit has the potential to control glycemic response. Our results show that the glycemic index of nutrient-extracted raspberry and mango (RM) and passionfruit and mango (PFM) were significantly lower than for the consumption of the whole fruits. Besides, the peak increment time after consuming tested meals were significantly longer after both nutrient-extracted RM and PFM than the consumption of the whole fruits. These results indicate that the nutrient extraction preparation of fruit with seeds could be a promising approach to reduce the glycemic response from consuming fruit in both healthy weight and in those individuals with obesity. 

Individuals with obesity, characterized as a BMI ≥ 30 kg/m^2^, are at increased risk of T2DM, and many individuals with obesity experience impaired glucose tolerance [9]. Globally, T2DM is considered as one of the most important health challenges; and many studies emphasize the positive association between the consumption of fruit juices and increased T2DM risk [10]. Our previous work demonstrated that homogenizing fruit using a ‘nutrient-extractor’ style blender, either improved, or did not worsen, the glycemic response amongst healthy, young adults, compared to eating the same fruit whole [6]. The purpose of the current work was to investigate whether our previous findings would also apply to individuals with obesity, who are at increased risk of developing T2DM, as it is important that general public health guidelines are applicable for the general population, of which 28.7% are individuals with obesity in the UK [5]. 

In keeping with our previous findings, the result of our current study, which was conducted with raspberries and mango, and passionfruit and mango, confirm that the consumption of these fruits in smoothie form resulted in significant improvements in the GI response in comparison to whole fruit. These findings challenge the current recommendations to limit smoothie consumption to 150 mL per day, and that they can only count towards one portion of fruit per day [11]. 

Our current study focused on combining two seeded fruits with mango, which is a high GI fruit. Our previous work focused on a mixed fruit smoothie (raspberries, passionfruit, kiwi(fruit), pineapple, mango and banana) and a mango smoothie. We observed a lowering of the GI response for the mixed fruit smoothie, compared with the same fruit consumed whole, and for mango, there was no difference in the GI response between mango consumed as a smoothie, or whole [6]. This present study found that participants experienced a significant reduction in GI levels after consuming the homogenized raspberries and passionfruit, both accompanied by mango, compared to eating the whole fruits. 

We postulate that this might be due to the technique of grinding the seeds during ‘nutrient extraction’, thus releasing fiber, polyphenols, fats and proteins, which can then be metabolized, thus slowing gastric emptying and the rate at which glucose is absorbed. We also hypothesize that it was perhaps these seeded fruits that were responsible for the lower GI observed for the mixed fruit smoothie in our previous work. 

Previous work has quantified the lipid, protein, carbohydrate, fiber and polyphenol composition of both passionfruit [12] and raspberry seeds [13], and importantly, the seed extracts and their by-products, from both fruits are considered safe for human consumption [14].

A previous study showed that consumption of a berry puree led to a significant decrease in peak glucose values compared to a control meal which consisted of sucrose, glucose and fructose in healthy participants [15]. This is in agreement with another study that emphasized the influence of polyphenolic components in raspberries on the inhibition of pancreatic α-amylase activity, which is an essential element to manage the blood glucose level in those with T2DM [16]. Additionally, other research has implied that polyphenols could positively modify the glucose utilization in mammals [17,18]. According to an analytical study of polyphenols, raspberry is one of the top 20 fruits that have a high concentration of polyphenols, nearly 600 mg/100 g per fruits [19]. 

This pattern of reduced glycemic response following the consumption of nutrient-extracted fruit was also observed for passionfruit. Passionfruit seeds contain a high proportion of piceatannol, which is a phenolic compound, and its extraction plays critical roles in inhibiting the glucose rise in the blood [20]. Passionfruit and piceatannol administration have shown a significant reduction in blood glucose levels in diabetic mice [21]. In addition, passionfruit fiber has the effect of controlling the glycemic index [22]. Therefore, these polyphenols have been shown to have benefical health effects and reduce the risk of chronic diseases, including diabetes. 

A potential limitation of our study is that the polyphenol contents, including piceatannol, were not measured in the fruit used in our study. However, results of other studies show that they differ according to their country of origin [23,24]. Another limitation, is that unlike our participants with obesity, who participated in both the raspberry-mango and passionfruit-mango arms of the study, each healthy weight participant participated in only one of the two arms. This was because our healthy weight participants were students who were not available for the second arm of the study due to time commitments, so we needed to recruit a second set of healthy weight participants. However, there were no statistically significant differences in baseline characteristics between the two sets of healthy participants, nor between the glucose values obtained at any time-point for the glucose control test meal, suggesting that it is valid to compare each group with those with obesity for each arm of the study. 

In keeping with previous studies, there was large inter-individual variation in the blood glucose values obtained at each timepoint for participants in both arms of the study, as demonstrated by large standard deviations [25,26]. This is a limitation of using GI to quantify the postprandial glucose response to food, which is widely recognized, and current research is examining the role of personalized nutrition with the use of algorithms to predict the glycemic response amongst individuals [27]. Additionally, our study only examined blood glucose responses following ingestion of the test meals. It would have been interesting to also examine insulin responses and changes in other hormones, such as Gastric Inhibitory polypeptide (GIP) or Glucagon-like peptide-1 (GLP-1) following ingestion of the meals.

For future work, the authors suggest isolating the seed fraction from the raspberries and passionfruit before blending, and assessing the glycemic index. This might give a useful insight into the role of seeds that could potentially be used as additives to other food/drinks to lower the GI.

In summary, our study demonstrates that fruit juice prepared using ‘nutrient-extractor’-style blenders elicits a lower glycemic response than the same fruit eaten whole in both healthy weight and individuals with obesity. Our findings challenge the current UK guidelines that smoothies should be treated the same as fruit juice, and should be restricted to 150 mL per day, and count towards just one portion of fruit. 

It is clear that many of the general public are consuming fruit in this way, and we suggest that larger studies examining different combinations of fruits, vegetables, nuts, milks and other ingredients commonly added to smoothies are conducted in large and diverse groups. Our study has shown the potential for ‘nutrient extracted’ fruit to elicit a more favorable glycemic response than whole fruit, even amongst individuals with obesity. Future studies should explore how the consumption of these smoothies may impact the blood glucose control of those with T2DM. 

## Figures and Tables

**Figure 1 nutrients-12-00766-f001:**
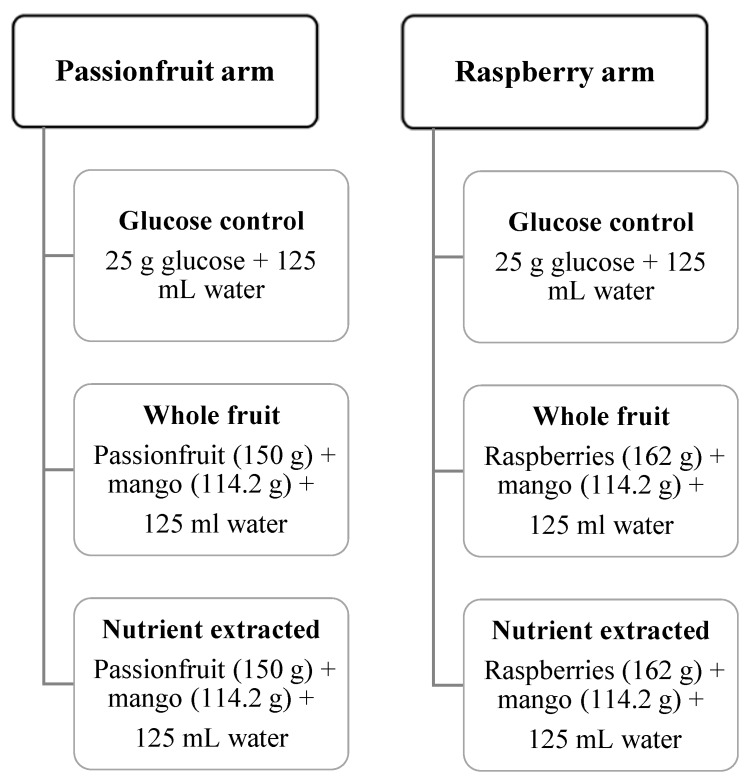
Schedule of planned experiments. Tests completed in random order. Each meal contained 25 g of total sugar.

**Figure 2 nutrients-12-00766-f002:**
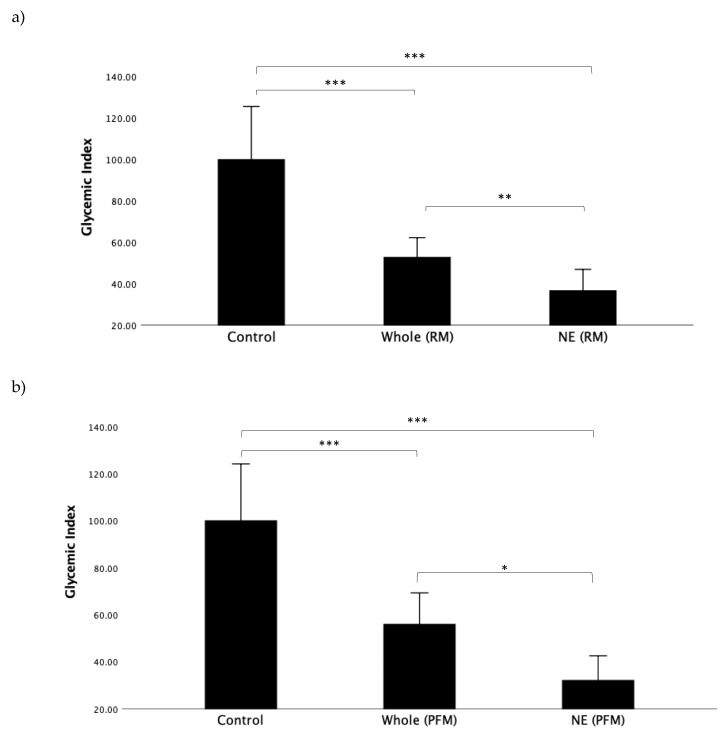
Comparison of glycemic index (GI) for each test meal: (**a**) raspberry and mango arm; (**b**) passionfruit and mango arm. * LSD post hoc test indicates significant different between all test * *p* value < 0.05, ** *p* ≤ 0.01, *** *p* ≤ 0.001.

**Figure 3 nutrients-12-00766-f003:**
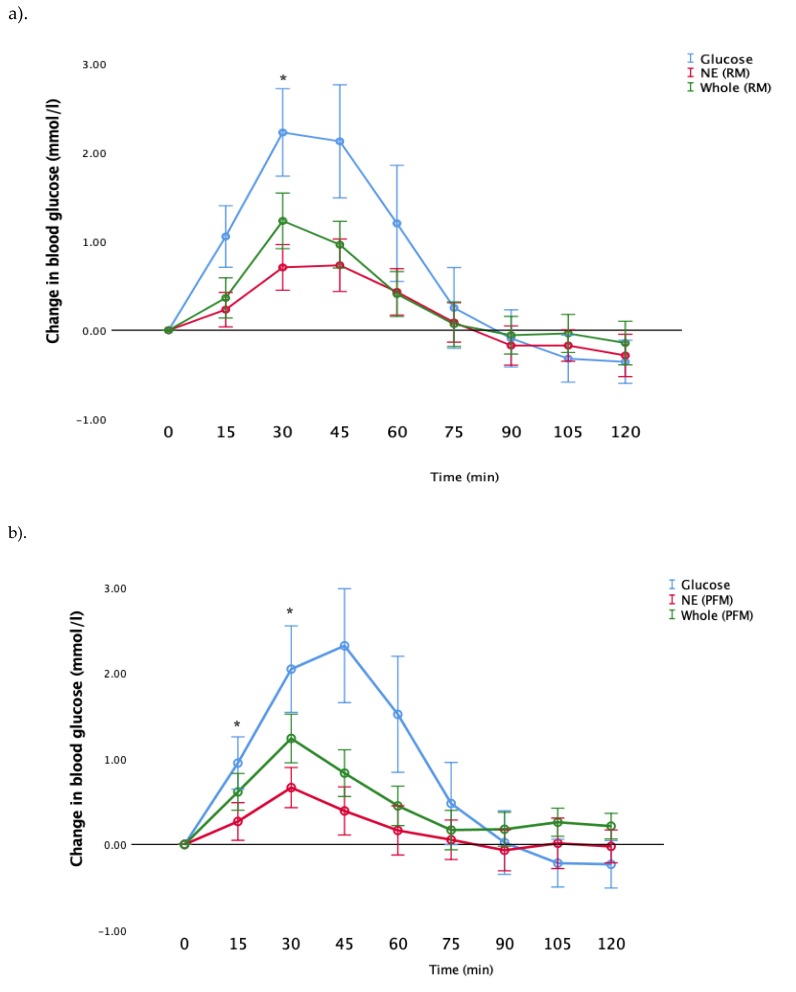
Comparison of incremental area under the curve values of blood glucose level after ingestion of the glucose control, whole and nutrient extracted (NE) fruit; (**a**) raspberry-mango arm; (**b**) passionfruit-mango arm. * LSD post hoc test indicates significant different between NE fruit and the whole *p* < 0.05.

**Figure 4 nutrients-12-00766-f004:**
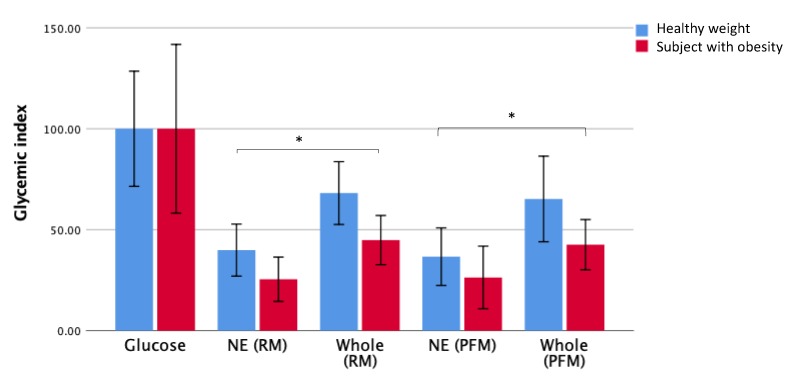
Comparison of GI for each test meal for groups with a healthy weight and obesity. NE = nutrient extracted; RM = raspberry/mango: PFM = passionfruit/mango * indicates a significant difference between NE and whole fruits in both individuals with healthy weight and obesity, *p* < 0.05.

**Figure 5 nutrients-12-00766-f005:**
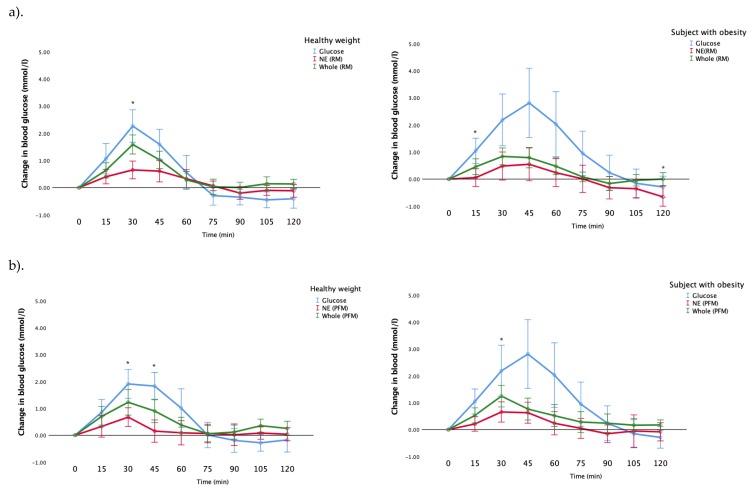
Comparison of the incremental changes in blood glucose values from baseline between individuals with healthy weight and obesity for each test meal. (**a**) raspberry-mango arm; (**b**) passionfruit -mango arm. * indicates the difference between the nutrient-extracted fruits and whole fruit *p* < 0.05.

**Table 1 nutrients-12-00766-t001:** Participant characteristics for each group. Presented as the mean ± standard deviation.

Participant	Gender		Age (years)	Weight (kg)	Height (cm)	BMI (kg/m^2^)	Body Fat (%)	WC (cm)	FBG (mmol/L)	Systolic BP (mmHg)	Diastolic BP (mmHg)	Cholesterol (mg/dL)
	M	F										
Healthy weight RM ^1^	6	9	27.00 *	64.81 *	170.77	22.10 *	24.14 *	76.44 *	5.07 **	112.50	74.75	202.62
n = 15			± 5.95	± 9.56	± 8.91	± 1.56	±7.70	± 12.21	± 0.56	± 13.59	± 5.41	± 69.37
Healthy weight PFM ^1^	4	6	31.87	63.75 *	170.75	21.82 *	24.35 *	76.63 *	4.78 ***	109.25	70.75	202.25
n = 10			± 9.99	± 7.57	± 7.32	± 1.92	± 9.91	± 11.56	± 0.44	± 15.64	± 9.34	± 68.90
Obese (both RM -PFM)	5	6	44.73	102.66	161.18	34.33	38.61	102.09	5.32	119.00	74.44	206.09
n = 11			± 10.71	± 22.72	± 26.11	± 2.96	± 7.70	± 11.97	± 0.55	± 14.69	± 13.94	± 63.78

^1^ Two groups of healthy weight subjects participated – one group for the raspberry and mango arm (RM), and one for the passionfruit and mango (PFM) arm. LSD post hoc test indicates HW groups were significantly different to the group with obesity: * *p* value <0.05, ** *p* ≤ 0.01, *** *p* ≤ 0.001. No significant differences were found between HW groups. WC = Waist Circumference, FBG = Fasting blood glucose.

**Table 2 nutrients-12-00766-t002:** Mean glycemic index ± standard deviation for raspberry-mango (RM).

Participant	Control (RM)	Whole (RM)	Nutrient-Extracted (RM)	*p*-Value between Meals within the Groups
Obese *n* = 11	100 ± 69.33	44.85 * ± 20.18	25.43 ± 18.20	< 0.05
Healthy weight *n* = 15	100 ± 35.62	68.13 ± 30.13	39.89 ± 24.97	< 0.05

* *t*-test indicates the significant difference between healthy weight and obese groups with *p* value = 0.036.

**Table 3 nutrients-12-00766-t003:** Mean glycemic index ± standard deviation for passionfruit-mango (PFM).

Participant	Control (PFM)	Whole (PFM)	Nutrient-Extracted (PFM)	*p*-Value between Meals within the Groups
Obese	100	42.56	26.30	<0.05
*n* = 11	± 69.33	± 20.64	± 25.72	
Healthy weight	100.55	73.49	30.99	<0.001
*n* = 10	± 34.49	± 39.14	± 12.11	

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
