# Peer review of "Nutrient Extraction Lowers Postprandial Glucose Response of Fruit in Adults with Obesity as well as Healthy Weight Adults"

_nutrients, 2020, doi:10.3390/nu12030766_

Round 1

Reviewer 1 Report

The paper examined the effect of different modes of fruit ingestion on GI in healthy and obese subjects. Only two mixes were examined, which severely limited the significance/impact of the paper. The only results reported were the blood glucose curves, which were then used to calculate the GI. The paper could also have been improved by examining other characteristics - insulin response, changes in gut factors such as GIP etc.

'Postprandial GI' should be phrased differently - GI is by definition the postprandial blood glycaemic change following a meal, so should be either the nutrient extraction had a lower 'GI' or lowered 'postprandial glycaemia\ reduced postprandial hyperglycaemia'.

[Line 10] Possibly explicate increased type 2 diabetes risk

[Line 27] states fruit juice is high GI, but the rest of the paper seems to focus on whole fruit (whose GI is not commented on) instead of juice?

What is the point in re-examining the data collectively versus weight group? The differences between the weight groups is not quantified, only within groups.

Author Response

Thank you so much for your constructive comments. Please see attachment for our response.

Reviewer 2 Report

In this paper, Alkutbe et al. evaluated the postprandial response of two different fruit preparations administered in two different forms, e.g. whole fruits and nutrient extract.

In my opinion, the study is of great interest, since it adds evidence to the debate on the between-meal fruit consumption in subjects at risk of developing insulin resistance.

The methodology is solid and the conclusion are supported by the results.

  • My main concern is about the presentation of the methods. At a first reading of Sections 2.1, 2.2, 2.3, the study design is a bit unclear. It takes further reading to understand who consumed what. To this regard, table 1 is not so explicative and very important information is reported in its footnote. I suggest replacing the table with a more informative diagram providing also a graphic representation of the crossover design. Moreover, the information on Table 1 footnote should be reported somewhere into the main text.
  • Table 2 should report mean fasting glucose for each group.
  • In the results, the number of subjects who have completed the study is reported. I suggest reporting the total number of recruited subjects in the methods, along with the reasons accounting for dropouts.
  • In figure 1, the authors should add significance for each pairwise comparison, and report a number of asterisks corresponding to the actual p value (e.g. *** for p<0.001)

Author Response

Thank you very much for your helpful comments. Please see our responses in the attached document.

Reviewer 3 Report

The article presents interesting research results but requires some addition. The presented research is particularly important in the context of the increasing incidence i of civilization diseases. Developing research undertaken by the authors is of great social importance. However, some issues need to be clarified.

Here are some small comments that will help to improve the  work:

In the methodology, please describe the age of the respondents more accurately. The information in the Table 2 does not exactly show this. Please enter the number of  participants in the survey in the methodology.

Did the different number of  participants  in each variant of the experiment affect the results obtained? Please discuss this issue using references to relevant  literature.

Why different scale was used in Figures 1a and 1b.

In the presented results, standard deviations amount to  several dozen percent. How to explain the high standard deviations of the obtained results. Please discuss this issue using references to relevant literature.

In many cases, the seeds are richer in polyphenols and other compounds compared to other parts of the fruit, such as pulp or skin. During grinding, these compounds can be transferred to the product. Could this have affected the results obtained? Please discuss this issue using references to relevant literature. Is there not a risk of unwanted substances passing from the seeds to the product? This is particularly important for the planning of further research by the authors.

Author Response

Thank you very much for reviewing our paper and for your helpful comments. Please see the attached document for our responses. 

Round 2

Reviewer 3 Report

Corrections made by the authors are sufficient. The article in its current form is suitable for printing.